# Analysis of Surface Topography Changes during Friction Testing in Cold Metal Forming of DC03 Steel Samples

**Tomasz Trzepieciński** [1] , **Krzysztof Szwajka** [2,*] **and Marek Szewczyk** [2]

1 Department of Manufacturing Processes and Production Engineering, Rzeszow University of Technology, al. Powst. Warszawy 8, 35-959 Rzeszów, Poland; tomtrz@prz.edu.pl
2 Department of Integrated Design and Tribology Systems, Faculty of Mechanics and Technology, Rzeszow University of Technology, ul. Kwiatkowskiego 4, 37-450 Stalowa Wola, Poland; m.szewczyk@prz.edu.pl
* Correspondence: kszwajka@prz.edu.pl

**Abstract:** Predicting changes in the surface roughness caused by friction allows the quality of the product and the suitability of the surface for final treatments of varnishing or painting to be assessed. The results of changes in the surface roughness of DC03 steel sheets after friction testing are presented in this paper. Strip drawing tests with a flat die and forced oil pressure lubrication were carried out. The experiments were conducted under various contact pressures and lubricant pressures, and lubrication was carried out using various oils intended for deep-drawing operations. Multilayer perceptrons (MLPs) were used to find relationships between friction process parameters and other parameters (Sa, Ssk and Sku). The following statistical measures of contact force were used as inputs in MLPs: the average value of contact force, standard deviation, kurtosis and skewness. Many analyses were carried out in order to find the best network. It was found that the lubricant pressure and lubricant viscosity most significantly affected the value of the roughness parameter, Sa, of the sheet metal after the friction process. Increasing the lubricant pressure reduced the average roughness parameter (Sa). In contrast, skewness (Ssk) increased with increasing lubrication pressure. The kurtosis (Sku) of the sheet surface after the friction process was the most affected by the value of contact force and lubricant pressure.

**Keywords:** surface roughness; sheet metal forming; steel sheets; surface topography

## 1. Introduction

The factors influencing the friction in conventional sheet metal forming using stamping dies include, among others, the topography of the cooperating surfaces, the character of the movement of the tools (static or dynamic) and physico-chemical phenomena in the contact interface and forming temperature [1]. The friction phenomenon occurring in hot and warm forming conditions is more intense than in cold forming conditions, owing to the intensification of the galling phenomenon. At high temperatures, the tribological system is subjected to extreme conditions, and no lubrication can be applied [2]. Physico-chemical phenomena occurring in the contact zone also depend on the materials of the rubbing pair. Large deformation values of the surface asperities and the related increased local temperature in micro-areas cause the phenomenon of adhesion [3,4].

Surface topography is used for the quantitative characterisation of sheet metal surfaces in metal forming [5,6]. Appropriate surface topography determines the formation of oil pockets [7], which reduce friction by creating a pressure lubricant cushion and acting as a lubricant reservoir. Closed lubricant pockets [8] are of the greatest importance in reducing the coefficient of friction. The lubricant contained in these pockets is subjected to high hydrodynamic pressure [9]. The efficiency of transferring the external load through the lubricant depends primarily on the viscosity of the lubricant and the load-bearing capacity of the sheet profile determined by the Abbott–Firestone curve [10].

These contact conditions affect the change in the initial topography of the sheet metal in the sheet metal forming processes [11]. Therefore, surface roughness must be considered as a variable factor during the forming process. It should also be noted that the real contact area, $A_{real}$, in the micro-scale is smaller than the nominal contact area, $A_{nom}$; hence, the load is transferred through the deformed summits of the asperities [12].

It is necessary to understand the impact of the surface roughness parameters of the tool and sheet metal on the phenomenon of friction. The kurtosis, Sku; the skewness, Ssk; the Sa and the Sq are the parameters that are used most often to describe the surface roughness of sheets in industrial practice [13]. However, the parameters Sq and Sq are strongly correlated, so analysing only one of these two parameters is sufficient. Sedlacek et al. [14] found that in the case of friction in the lubrication regime, Sku and Ssk are the most suitable for describing the friction occurring during conventional sheet metal forming. Surfaces with more negative values of the skewness parameter and with higher kurtosis values show lower values of the friction coefficient and, consequently, change to a lesser extent during the metal forming process.

The study of changes in surface topography has been the occupation of many authors in recent years. Wu et al. [15] predicted the effect of surface roughing based on the material data of DX56 low-carbon steel sheet metal. They found that surface deformation has a negative effect on the lubrication in metal forming. Azushima and Kudo [16] observed a sheet surface in a friction test and concluded that the flattening of the surface asperities influenced the value of the friction coefficient. Han et al. [17] established a finite element-based model of surface topography based on analysing the effect of forming parameters on the surface topography of T3 pure copper sheet metal. It was concluded that the surface roughness of the sheet metal depended approximately linearly on the initial surface roughness of the workpiece. Klimczak et al. [18], based on the results of analytical studies, recommended the Sq parameter for controlling the sheet metal-formed draw-pieces in production. Sigvant et al. [19] numerically investigated the stamping die surface roughness in a VDA239 CR4 GI sheet material. The increase in the blankholder surface roughness resulted in significantly higher restraining of the workpiece. Consequently, the coefficient of friction increased. Sanchez and Hartfield-Wunsch [20] analysed the evolution of surface roughness changes on mill-finished and electron discharge textured (EDTed) aluminium sheets using a draw dead simulator. Average surface roughness parameter, Sa, changes of up to 400% were observed for the mill-finished material. In the case of EDT surfaces, the maximum charge of the roughness parameter, Sa, was approximately 30%. Trzepieciński et al. [21] investigated the evolution of the surface topography due to the deformation of a DC04 steel sheet in a friction test with rounded countersamples. It was found that an increase in the plastic deformation of sheets caused an increase in the value of parameters Rp, Ra and Rt, measured both along the rolling direction of the sheet metal and across it. Zhang et al. [22] developed a finite element (FE) model of the contact of a single roughness peak and flat tool. They concluded that the height of the asperity decreased with increased local pressure, and this effect was much higher than the nominal pressure. Çavuşoğlu and Gürün [23] investigated the surface roughness effect on the formation of the EN AW-3003-H111 sheets using the FE method. The surface roughness influenced the amount of sheet thinning (influence of 56%) in the process of stretching the sheet with a hemispherical punch.

According to this literature review, most authors are focused on the experimental or numerical investigation of the effect of test parameters on the surface topography of deformed sheets. It should be emphasised that the experimental determination of the intricate relations between many friction parameters and the surface roughness of a workpiece is complex. For this reason, in this paper, artificial neural networks (ANNs) were employed to find the relation between parameters of the friction process and the main surface roughness parameters. The advantage of artificial neural networks is that they provide ability to acquire knowledge, even on a limited set of data, as a result of the training process. The mean roughness, Sa, most used in the industrial practice of sheet

metal forming, was selected as the main roughness parameter. Based on the results of Sedlacek et al. [14], the skewness, Ssk, and the kurtosis, Sku, were also considered.

## 2. Methods and Materials

### 2.1. Test Material

The workpiece material was a DC03 (1.0347) steel sheet with a thickness of 1.2 mm. Mild low-carbon DC03 steel exhibits excellent deep-drawing capability. The chemical constitution of this steel is shown in Table 1. For information purposes, the mechanical properties of this sheet were determined using samples cut along the rolling direction. Uniaxial tensile tests on a universal testing machine were carried out in accordance with standard EN ISO 6892-1 [24]. The average values of the selected mechanical parameters (yield stress $R_{p0.2}$, ultimate tensile strength $R_m$ and elongation $A_{50}$) based on three measurements are presented in Table 2.

**Table 1.** Chemical constitution of the DC03 material (wt.%).

| C | P (max.) | Mn | S | Al | N | Fe |
|---|---|---|---|---|---|---|
| 0.05 | 0.20 | 0.20 | 0.01 | 0.04 | 0.003 | remainder |

**Table 2.** Basic material parameters of the DC03 sheet.

| $R_{p0.2}$, MPa | $R_m$, MPa | $A_{50}$, % |
|---|---|---|
| 203.9 | 322.7 | 23.9 |

### 2.2. Experimental Methodology

In the experimental studies, the basic and the most used test to determine the value of the friction coefficient of was used. Formally, this friction test is used to determine the friction coefficient in the region of action of the blankholder in metal forming. The test was conducted using a specially designed device (Figure 1a), allowing for innovative pressure-assisted lubrication. The device was mounted in the lower holder of the universal testing machine. Lubricant was supplied to the contact zone through channels (Figure 1b) connected to an Argo-Hytos hydraulic pump (Figure 2).

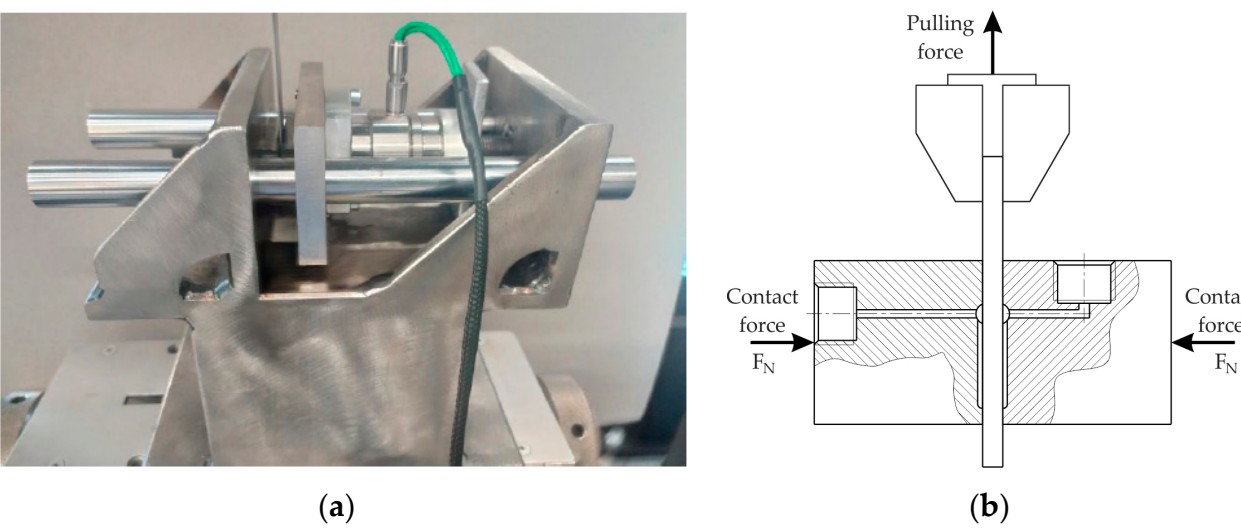

<div align="center">(<b>a</b>)         (<b>b</b>)</div>

**Figure 1.** (**a**) View of the tribotester and (**b**) the cross-sectional view of the working countersamples.

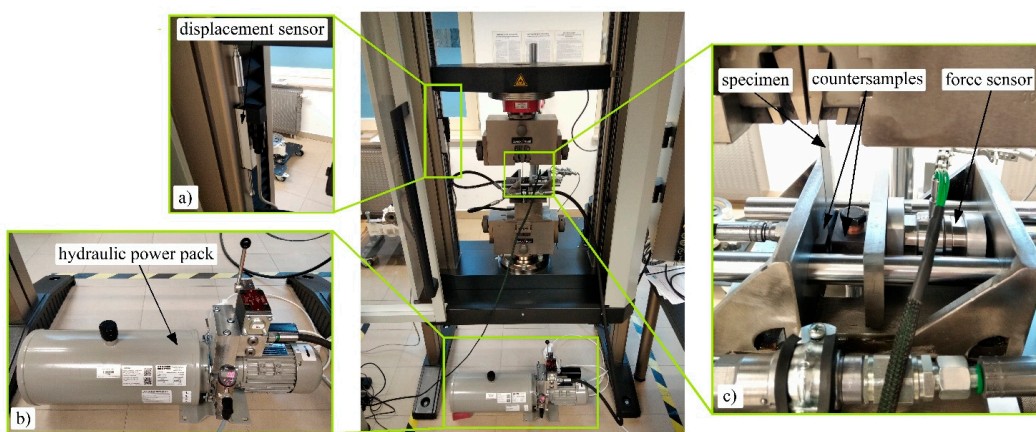

**Figure 2.** Experimental stand: (**a**) strain sensor, (**b**) hydraulic power pack, (**c**) tribotester [25].

The working part of the device consisted of two countersamples with a flat surface made of 145Cr6 cold work steel (hardness of 197.2 HV). The test involved pulling strips of sheet metal between countersamples pressed together with a specific force. The dimensions of the strips were as follows: width = 25 mm, and length = 130 mm. Various contact force $F_N$ (Figure 1b) values (1000, 2000, 3000 and 4000 N) were used to obtain nominal contact pressures, $p_c$, of 2, 4, 6 and 8 MPa. Two synthetic oils, intended specifically for lubricating sheet metals in deep-drawing operations—S100 Plus (kinematic viscosity at 20 °C: 360 mm$^2$/s) and S300 (kinematic viscosity at 20 °C: 1136 mm$^2$/s)—both produced by Naftochem, were used in the tests. The value of the contact force was set using a clamping screw. The force sensor Kistler type 9345B was used to control the value of the contact force. Oil was delivered to the contact zone conventionally, without pressure-assisted lubrication. In comparison, the lubricants were supplied at pressure, $p_l$, of 0.6, 1.2 and 1.8 MPa. The upper value of lubricant pressure was selected in such a way as not to lead to oil leakage from the contact zone. So, the value of 1.8 MPa was the maximum oil pressure without leakage under the conditions of the applied contact forces between 1000 and 4000 N. The strip drawing tests were carried out at 20 °C. The effect of the temperature on the friction coefficient determined in strip drawing tests at room temperature is generally ignored, especially during tests with low pressures and sliding velocities [4,26–28]. Some publications point out the increase in the temperature of the surface asperities due to friction [26]; however, measuring the temperature in these asperities is problematic. So, the research assumed that the effect of changing the temperature in the friction interface on the coefficient of friction was negligible, and each strip was tested in stable conditions.

*2.3. Surface Roughness*

The main three-dimensional surface roughness parameters of the as-received specimens and specimens after friction testing were determined on a surface of 5 × 5 mm using a T8000RC profilometer. The following surface roughness parameters were determined according to standard ISO 25178-2 [29]: Sa, Sq, Ssk, Sp, Sz and Sv.

The surface topography of the workpiece is shown in Figure 3. The surfaces of the countersamples were characterised by the following parameters: Sa = 0.237 μm; Sq = 0.384 μm; Sku = 24.7; Ssk = –2.87; Sp = 4.28 μm; Sz = 10.8 μm; Sv = 6.50 μm.

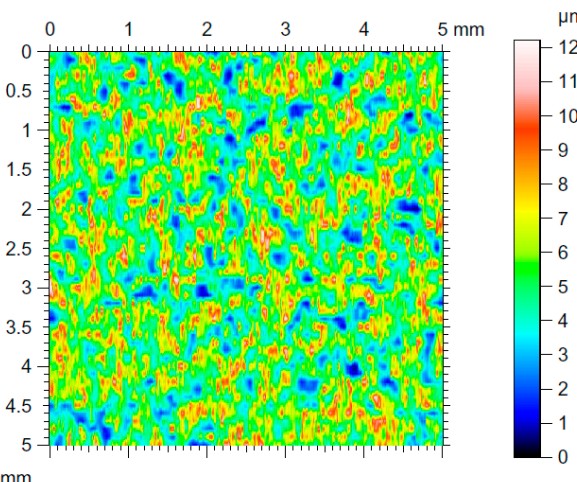

**Figure 3.** The topography of the DC03 steel sheet.

### 2.4. Determination of Measures from Contact Force Signal

For the analysis of the registered contact force ($F_N$) signals, our own application in the LabVIEW environment was developed for the analysis of the values of selected statistical measures of the registered contact force signals. The following statistical measures of contact force, $F_N$, were analysed: the average value of the contact force ($F_{Nmean}$), standard deviation ($F_{NSD}$), kurtosis ($F_{Nku}$) and skewness ($F_{Nsk}$). However, before it was possible to determine the values of the adopted statistical measures, it was necessary to select such fragments of the nominal force signal that best represented the value of the contact force, $F_N$.

The signal fragments from the stabilised waveform were the most suitable for such analysis. They avoided random changes in the value of the contact force signal. Before proceeding to determine the adopted measures of the force signal, the signal offset had to be removed. After removing the offset, the search for the most useful fragment of the signal began. After determining the beginning of the friction process, signal segmentation was carried out, which consisted of dividing the contact force signal (during its analysis) into equal time fragments. Then, from such segments, the average value of the analysed signal was determined. The method of evaluating the stability of the signal (fluctuation) has been presented and described in detail in [30]. Statistical measures from the contact force signal were determined from a fragment of the signal corresponding to the displacement of the sheet in the process over a length of 20 mm (Figure 4).

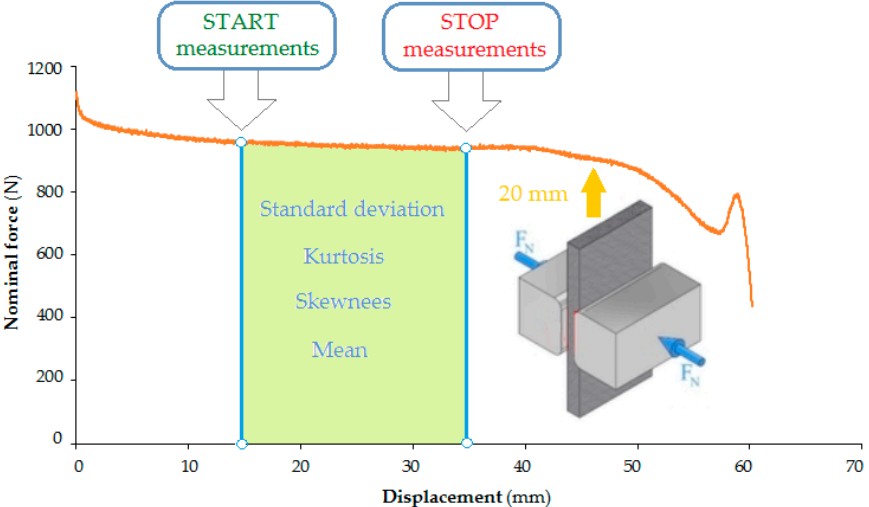

**Figure 4.** Method for determining statistical measures from the $F_N$ signal.

### 2.5. Artificial Neural Networks

The analyses with the multilayer perceptrons (MLPs) were carried out based on the results of the friction test. As numerical input parameters, oil pressure and measures of contact force signals listed in Section 2.4 ($F_{Nmean}$, $F_{NSD}$, $F_{Nku}$, $F_{Nsk}$) were selected. Friction conditions were considered at the network input as categorisation variables: no friction, conventional lubrication (without pressure-assisted lubrication) and pressure-assisted lubrication with oil pressure $p_l$ of 0.6, 1.2 and 1.8 MPa. At a later stage, the significance of these variables was checked by means of a sensitivity analysis. The variation in the input parameters and coefficient of friction, $\mu$ (output parameter), is presented in Table 3.

**Table 3.** The variation in input parameters and coefficient of friction.

| Parameter | Range of Variability |
|:---:|:---:|
| $F_N$, kN | 1–4 |
| $p_C$, MPa | 2–8 |
| $p_l$, MPa | 0–1.8 |
| $F_{Nmean}$, N | 960.4–4359.9 |
| $F_{NSD}$, N | 2.13–12.24 |
| $F_{Nku}$ | −1.18–0.53 |
| $F_{Nsk}$ | −0.41–1.01 |
| $\mu$ | 0.069–0.341 |
| Sa, $\mu$m | 0.982–1.64 |
| Sku | 2.63–3.18 |
| Ssk | −0.0945–0.498 |

The training set contained 36 experimental training sets (input parameters and a corresponding output value).

Most attention was paid to the multilayer perceptrons (MLPs) and their prediction possibilities in terms of the surface roughness of the specimens after the friction process. Unidirectional MLPs are the most frequently described and the most often used neural network architectures in practical applications. Their dissemination is related to the development of the back propagation training algorithm, which enabled effective training of this type of network in a relatively simple way.

The purpose of this analysis was to find the optimal structure of the MLP and type of activation function while maintaining a good quality of response, i.e., a low number of errors. To model the considered problem using MLPs, the Statistica 13.3 program was used. The minimisation of the network structure mainly involved the size of the input vector—the number of measures and friction parameters analysed by the network. The number of hidden neurons was optimised. The input variables were oil pressure, lubricant viscosity and selected statistical measures of the contact force, $F_N$ (mean value, standard deviation, skewness, kurtosis). On the other hand, the surface roughness parameters, Sa, Ssk and Sku, were used as the output. The input and output parameters used in the selection of the ANN architecture are presented in Figure 5.

In addition, when choosing the appropriate architecture of the ANN, the adopted type of the activation function was also considered (Figure 6).

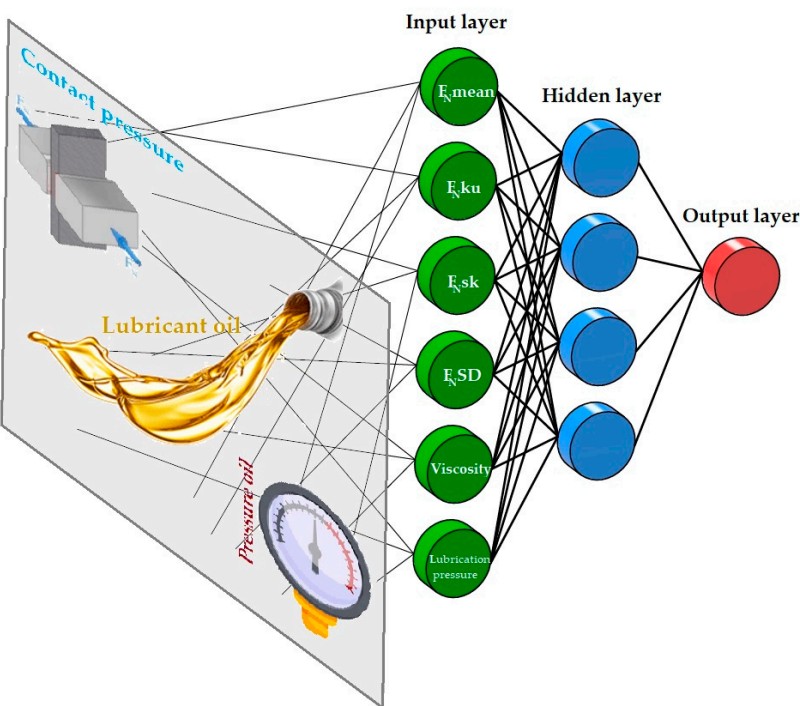

**Figure 5.** The structure of the ANN adopted in the analyses.

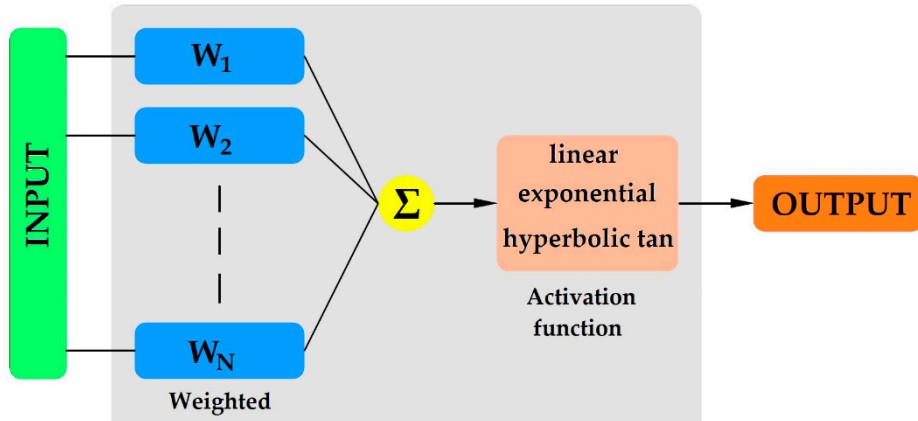

**Figure 6.** Structure of the MLP.

Three types of the neuron activation function were used for the analysis: linear, exponential and hyperbolic. Their general function of activation functions is to map any real input signal over a limited range, typically 0 to +1 or −1 to +1. In this study, three of the most used activation functions were selected to investigate their effect on the prediction of the surface topography of sheet metal using MLPs.

The operation of the artificial neural network when choosing a linear function consists of directly transferring the value expressing the total activation of the neuron to its output. The problem with a linear activation function is that it cannot be defined within a certain range. It has a range $(-\infty, \infty)$, and this makes the ANN sometimes unable to cope with complex problems. Another limitation of ANN application is that the gradient does not depend on the inputs at all. As a result, during backpropagation, the error changes in a constant way. Regardless of the structure of MLP, the last layer always acts as a linear function of the first layer, which makes it difficult to analyse complicated problems.

The exponential and Gaussian-like activation functions are perfect for use in radial basis neural networks [31,32]. However, they are also commonly used in MLPs [33]. The

function outputs, $y$, are from 0 to $\infty$. The combination of the radial aggregation function and the exponential function with a negative exponent de facto defines the neurons modelling the Gaussian function centred in relation to the weights vector [34]:

$$y = e^{-\alpha} \tag{1}$$

A hyperbolic function (Equation (2)) is an S-shaped (sigmoidal) function which is commonly used in MLPs. The hyperbolic activation function is often approximated by the mathematical function hyperbolic tangent (denoted tanh or tgh). Like the logistic (sigmoid) function, it is an S-shaped curve [34]. Such a function can work perfectly well in numerous non-linear neural networks—especially in multi-layer perceptrons [35,36].

$$y = \frac{1}{1 - e^{-\alpha}} \tag{2}$$

## 3. Results and Discussion

### 3.1. MLP Architectures

From the set of experimental data, the test and validation sets of data, whose task was to check the correct operation of the training algorithm, were separated, and the rest of the data were included in the so-called training set (used directly for teaching the network). The experimental set of data was divided into three groups: training data, validation data and test data in proportions of 70%, 15% and 15%, respectively.

Some surface roughness parameters can be correlated in the specific conditions of the surface topography; therefore, it was assumed necessary to consider independent multilayer networks with only one of the roughness parameters, Sa, Ssk or Sku, at the output. Then, the results of the best networks from each group of input parameters were independently evaluated. In the experiments regarding the search for optimal MLP models, the Statistica 13.3 package was used. It was assumed that the network designer build in Statistica software would generate a certain number of networks that would be trained based on the available data from experiments and that only the best three networks would be retained after validation of the training results.

A search was carried out for the most suitable network architecture. The predictive accuracy of neural networks is subject to random effects, and even ANNs with different architectures may have similar predictive properties due to a more favourable initial set of weights. On this basis, analyses were carried out for networks with different architectures and different numbers of neurons. Three networks were selected to predict each of the selected surface roughness parameters of sheet metal. Then, based on the training error values for the validation and training sets, the most advantageous network was selected. The architectures of three networks with the smallest errors for each of the output parameters and each neuron activation function are presented in Tables 4–12 (in the second column), and in the further columns, the values of quality parameters of the MLPs are listed. In these tables, the training, test and validation sets are abbreviated as TR, TE and V, respectively.

**Table 4.** Quality parameters of the MLP with a linear activation function for the prediction of the Sa parameter.

| No. | MLP Architecture | Correlation | | | Error | | |
|---|---|---|---|---|---|---|---|
| | | TR | TE | V | TR | TE | V |
| 1 | MLP 10-6-1 | 0.848561 | 0.799852 | 0.456798 | 0.006260 | 0.005392 | 0.007145 |
| 2 | MLP 10-9-1 | 0.848783 | 0.790019 | 0.457024 | 0.006246 | 0.005529 | 0.007192 |
| 3 | MLP 10-7-1 | 0.848783 | 0.790019 | 0.457024 | 0.006246 | 0.005529 | 0.007192 |

**Table 5.** Quality parameters of the MLP with the exponential activation function for the prediction of the Sa parameter.

| No. | MLP Architecture | Correlation | | | Error | | |
|---|---|---|---|---|---|---|---|
| | | TR | TE | V | TR | TE | V |
| 1 | MLP 10-5-1 | 0.964086 | 0.966114 | 0.962563 | 0.001618 | 0.001729 | 0.001357 |
| 2 | MLP 10-10-1 | 0.969939 | 0.977829 | 0.943718 | 0.001385 | 0.000794 | 0.002179 |
| 3 | MLP 10-4-1 | 0.965439 | 0.991373 | 0.971247 | 0.001553 | 0.000574 | 0.000816 |

**Table 6.** Quality parameters of the MLP with the tanh activation function for the prediction of the Sa parameter.

| No. | MLP Architecture | Correlation | | | Error | | |
|---|---|---|---|---|---|---|---|
| | | TR | TE | V | TR | TE | V |
| 1 | MLP 10-6-1 | 0.961718 | 0.961872 | 0.936167 | 0.001681 | 0.001342 | 0.001421 |
| 2 | MLP 10-11-1 | 0.949534 | 0.989222 | 0.931313 | 0.002200 | 0.001625 | 0.001367 |
| 3 | MLP 10-7-1 | 0.980119 | 0.990977 | 0.949637 | 0.000880 | 0.001272 | 0.000859 |

**Table 7.** Quality parameters of the MLP with a linear activation function for the prediction of the Ssk parameter.

| No. | MLP Architecture | Correlation | | | Error | | |
|---|---|---|---|---|---|---|---|
| | | TR | TE | V | TR | TE | V |
| 1 | MLP 10-8-1 | 0.642631 | 0.234950 | 0.754865 | 0.004041 | 0.018242 | 0.007486 |
| 2 | MLP 10-4-1 | 0.657397 | 0.232405 | 0.787614 | 0.003797 | 0.017681 | 0.005741 |
| 3 | MLP 10-6-1 | 0.614863 | 0.243348 | 0.756677 | 0.004418 | 0.016229 | 0.005429 |

**Table 8.** Quality parameters of the MLP with the exponential activation function for the prediction of the Ssk parameter.

| No. | MLP Architecture | Correlation | | | Error | | |
|---|---|---|---|---|---|---|---|
| | | TR | TE | V | TR | TE | V |
| 1 | MLP 10-4-1 | 0.686226 | 0.262450 | 0.780858 | 0.003634 | 0.017766 | 0.005997 |
| 2 | MLP 10-3-1 | 0.681380 | 0.255007 | 0.783729 | 0.003593 | 0.017347 | 0.005152 |
| 3 | MLP 10-6-1 | 0.655258 | 0.298071 | 0.844212 | 0.003844 | 0.016428 | 0.004866 |

**Table 9.** Quality parameters of the MLP with the tanh activation function for the prediction of the Ssk parameter.

| No. | MLP Architecture | Correlation | | | Error | | |
|---|---|---|---|---|---|---|---|
| | | TR | TE | V | TR | TE | V |
| 1 | MLP 10-7-1 | 0.632725 | 0.233232 | 0.869024 | 0.003965 | 0.018361 | 0.006852 |
| 2 | MLP 10-8-1 | 0.613228 | 0.407401 | 0.847233 | 0.004429 | 0.016867 | 0.005412 |
| 3 | MLP 10-9-1 | 0.659507 | 0.268853 | 0.818048 | 0.003788 | 0.017803 | 0.005567 |

**Table 10.** Quality parameters of the MLP with a linear activation function for the prediction of the Sku parameter.

| No. | MLP Architecture | Correlation | | | Error | | |
|---|---|---|---|---|---|---|---|
| | | TR | TE | V | TR | TE | V |
| 1 | MLP 10-8-1 | 0.891092 | 0.925749 | 0.392733 | 0.002065 | 0.004704 | 0.008775 |
| 2 | MLP 10-3-1 | 0.891206 | 0.920480 | 0.409767 | 0.002062 | 0.004854 | 0.008590 |
| 3 | MLP 10-4-1 | 0.889840 | 0.922791 | 0.489860 | 0.002144 | 0.004079 | 0.007595 |

**Table 11.** Quality parameters of the MLP with the exponential activation function for the prediction of the Sku parameter.

| No. | MLP Architecture | Correlation | | | Error | | |
|---|---|---|---|---|---|---|---|
| | | TR | TE | V | TR | TE | V |
| 1 | MLP 10-4-1 | 0.983367 | 0.940394 | 0.924044 | 0.000331 | 0.003057 | 0.002618 |
| 2 | MLP 10-3-1 | 0.967260 | 0.902881 | 0.923602 | 0.000650 | 0.003811 | 0.003660 |
| 3 | MLP 10-11-1 | 0.990302 | 0.969133 | 0.955919 | 0.000196 | 0.001395 | 0.001561 |

**Table 12.** Quality parameters of the MLP with the tanh activation function for the prediction of the Sku parameter.

| No. | MLP Architecture | Correlation | | | Error | | |
|---|---|---|---|---|---|---|---|
| | | TR | TE | V | TR | TE | V |
| 1 | MLP 10-9-1 | 0.993754 | 0.954898 | 0.837398 | 0.000129 | 0.003025 | 0.002892 |
| 2 | MLP 10-11-1 | 0.990686 | 0.952943 | 0.839915 | 0.000195 | 0.003630 | 0.003490 |
| 3 | MLP 10-5-1 | 0.976751 | 0.951792 | 0.880115 | 0.000470 | 0.003789 | 0.002757 |

### 3.2. MLP for Prediction of Roughness Parameter Sa

Considering the smallest training errors for both the training and validation sets, the MLP 10-7-1 network (Table 6) was selected for further analysis. For this network, the values of Pearson's correlation R (Equation (3)) for the training, test and validation sets were approximately 0.98, 0.99, and 0.94, respectively. The network training process was carried out until there was no further decrease in the network error value for the test set. Further continuation of the learning process, even with a continuous decrease in error for the test set, led to the excessive correlation of the network response to the training data. The ability of the MLP network to generalise was assessed on the data contained in the test set, the data of which were not used in the learning process. The training process was terminated after 32 epochs (Figure 7).

$$R = \frac{\sum_{i=1}^{n} \left( x_i - \frac{1}{n}\sum_{i=1}^{n} x_i \right) \left( y_i - \frac{1}{n}\sum_{i=1}^{n} y_i \right)}{\sqrt{\sum_{i=1}^{n} \left( x_i - \frac{1}{n}\sum_{i=1}^{n} x_i \right)^2} \sqrt{\sum_{i=1}^{n} \left( y_i - \frac{1}{n}\sum_{i=1}^{n} y_i \right)^2}} \tag{3}$$

where n is the sample size, and $x_i$ and $y_i$ are individual sample points indexed with 'i'.

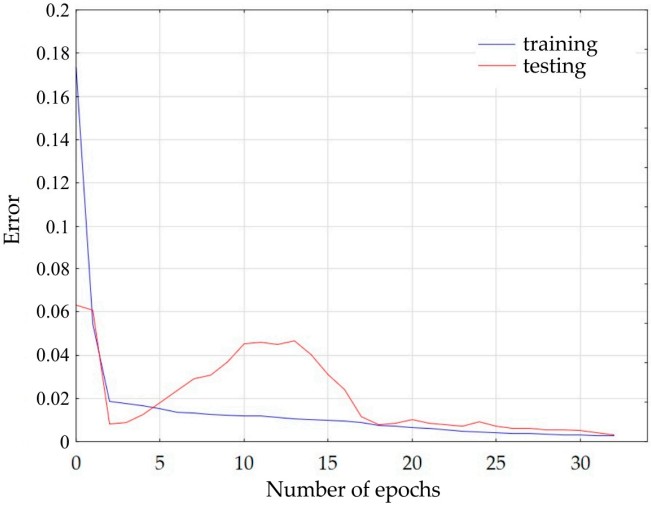

**Figure 7.** Variation in the network error during the training process.

Variable sensitivity analysis makes it possible to assess the significance of the influence of a specific variable on the value predicted by the output neuron. The basic measure of network sensitivity is the quotient of the error obtained for a data set that does not contain a specific variable and the error obtained with a complete set of variables [34]. The global sensitivity analysis for selected variables in the training set is as follows: oil pressure 57.811, oil viscosity 7.620, $F_{NSD}$ 4.377, $F_{Nku}$ 2.978, $F_{Nmean}$ 2.099 and $F_{Nsk}$ 1.277 (Table 13).

**Table 13.** Sensitivity analysis of input variables for MLP 10-7-1.

| Oil Pressure | Oil Viscosity | $F_{NSD}$ | $F_{Nku}$ | $F_{Nmean}$ | $F_{Nsk}$ |
|---|---|---|---|---|---|
| 57.811 | 7.620 | 4.377 | 2.978 | 2.099 | 1.277 |

If the error ratio presented in Table 13 is equal to or greater than 1, then this variable has a significant impact on the mean roughness parameter (Sa). The variables that affect the value of the Sa roughness parameter most are oil viscosity and its pressure. Subsequently, the surface roughness parameter, Sa, after the friction process is influenced by statistical measures of force, $F_N$: standard deviation ($F_{NSD}$), kurtosis ($F_{Nku}$), mean value of contact force ($F_{Nmean}$) and skewness ($F_{Nsk}$).

The effect of oil viscosity and oil pressure on the value of the Sa parameter should be considered together. The viscosity of the oil determines the durability of the lubricant film and its adhesion to the surfaces of the sheet and the countersample. When lubricating with higher-viscosity oil, higher pressures can be applied in a strip drawing test without the risk of lubricant leaks. In turn, low-viscosity oil is characterized by low flow resistance, and it is easier to break the lubricating film, even at low pressures. Under conditions of increasing contact pressure, the influence of lubricant pressure on changing the surface topography decreases. However, the mechanical cooperation of the surface asperities through flattening and ploughing mechanisms begins to play a dominant role.

Figures 8–11 show a comparison of measured data and responses of the MLP 10-7-1 network for selected configurations of input parameters. The response of the neural network (Figures 8b, 9b, 10b and 11b) perfectly reflects the results of experimental studies (Figures 8a, 9a, 10a and 11a). Under lubricated conditions, if the lubricant pressure increases, the Sa value decreases. This generally applies to negative values of the kurtosis of contact force. For positive values of kurtosis, the Sa value decreases with increasing lubricant pressure. Higher lubricant pressure ensures better penetration of the cavities located in the roughness valleys and, in the case of closed lubricant pockets, an additional reduction in the metallic interaction of the surface roughness asperities.

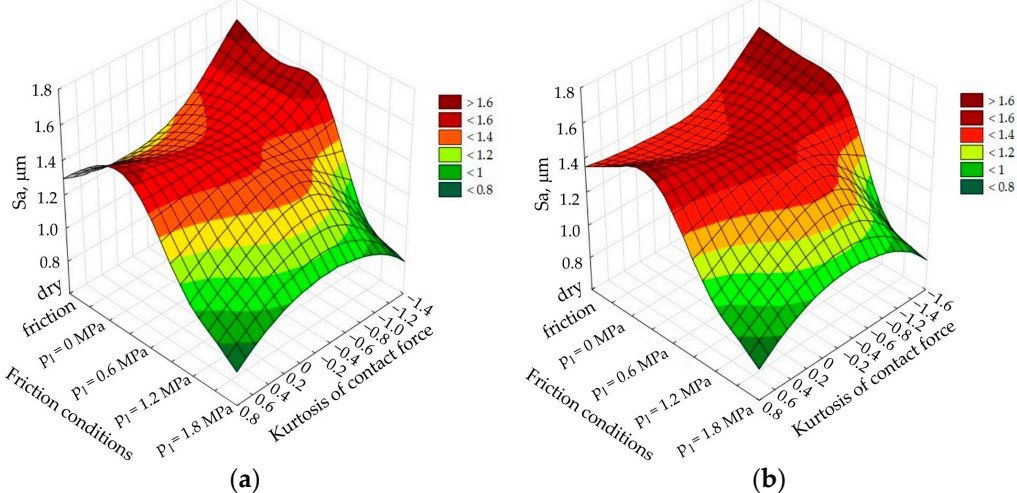

**Figure 8.** The effect of kurtosis of contact force and oil pressure on the value of Sa parameter: (**a**) measured data, (**b**) response surface for predicted data.

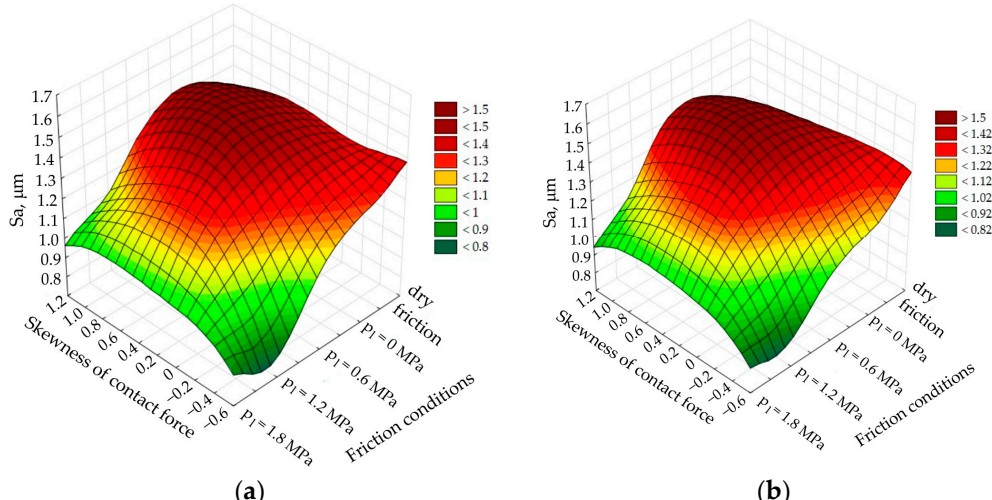

**Figure 9.** The effect of skewness of contact force and oil pressure on the value of Sa parameter: (**a**) measured data, (**b**) response surface for predicted data.

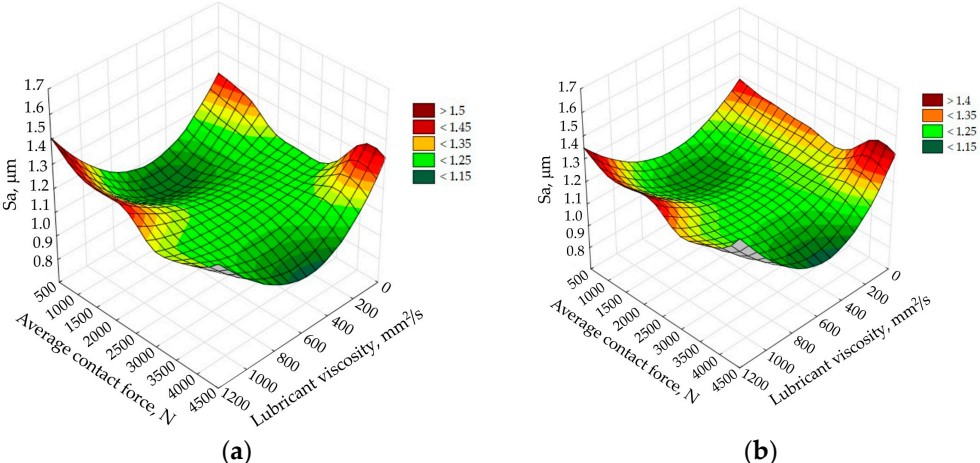

**Figure 10.** The effect of average contact force and oil viscosity on the value of the Sa parameter: (**a**) measured data, (**b**) response surface for predicted data.

The greatest increase in the value of the Sa parameter is observed for dry friction conditions and during conventional lubrication ($p_l = 0$ MPa). During lubrication with a lubricant pressure of $p_l = 1.8$ MPa, the smallest difference in the Sa parameter of the sheet metal after friction is observed in relation to the as-received sheet metal (Sa = 1.40 µm). Under these conditions, the mean surface roughness Sa increased by approximately 29%–43% (Figure 8). However, in dry friction conditions, the mean surface roughness varied by approximately 8%–21% depending on the kurtosis of contact force. In conventional lubrication conditions ($p_l = 0$ MPa), the mean roughness increased by approximately 2%–14%, depending on the kurtosis of contact force. So, the pressurised lubrication ensures lower average roughness of the workpiece surface after friction compared to the conventional lubrication. In [37], it was observed that friction under pressure lubrication conditions reduces the mean roughness of DC01 steel sheets by approximately 32%. Similarly, in [38], it was found that the Sa parameter of DC04 steel sheets increases with a reduction in normal pressure by approximately 12%–47%, in the range of contact pressures between 3 and 12 MPa. Most published works indicate a reduction in the surface roughness of sheet metal as a result of the surface flattening of asperities [39,40], or an increase in it as a result of the ploughing mechanism [41].

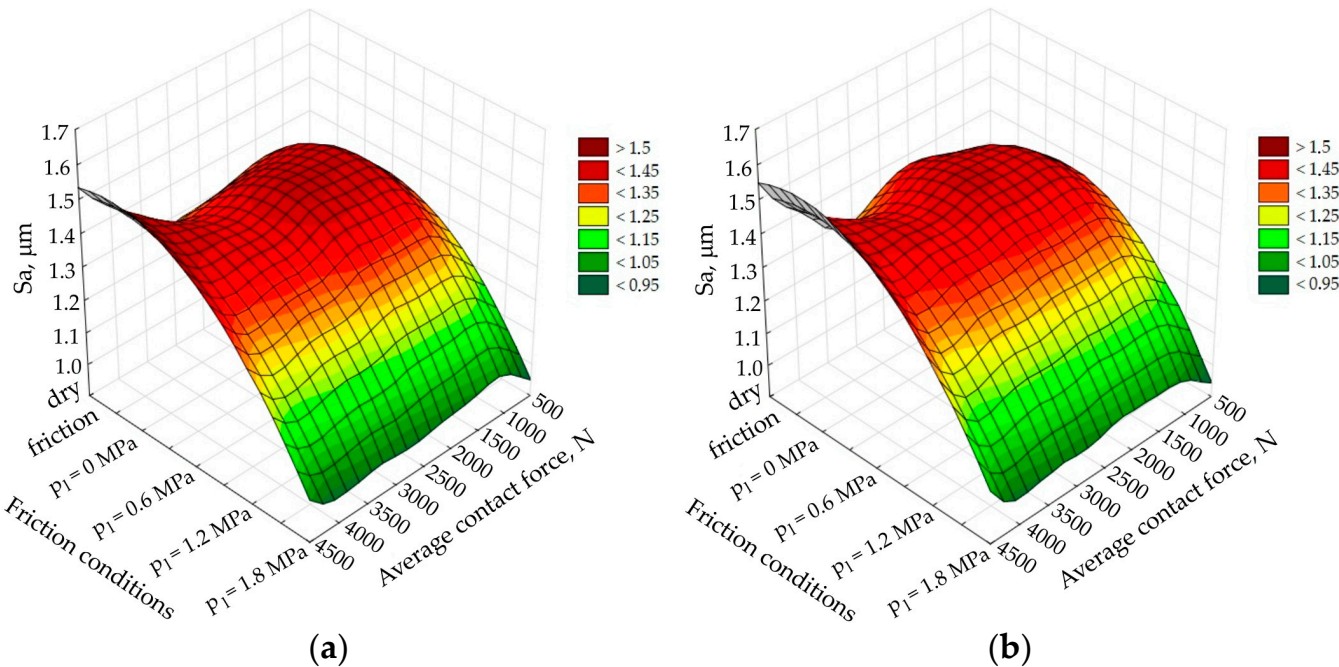

**Figure 11.** The effect of kurtosis of contact force and lubricant pressure on the value of the Sa parameter: (**a**) measured data, (**b**) response surface for predicted data.

The influence of the skewness of contact force and lubricant pressure on the roughness parameter, Sa, is similar (Figure 9). Increasing the oil pressure reduces the surface roughness of the sheet metal defined by the Sa parameter.

The response surface of the neural network for the influence of the average contact force and lubricant viscosity is represented by two troughs with the minimum values of the Sa parameter (Figure 10). The most unfavourable friction conditions due to the minimisation of the Sa roughness parameter occur at the highest average contact force and in the absence of lubrication (lubricant viscosity of 0 mm$^2$/s).

In the range of the average contact force between about 1500 and 3500 N, a stabilisation of the Sa roughness parameter value for specific lubrication conditions is visible (Figure 11). Irrespective of the value of the oil pressure, first, the Sa parameter increases; then, its stabilisation occurs in a wide range of changes described above, and finally, after exceeding the value of the average contact force of 3500 N, the Sa parameter increases sharply. This may be justified by the intensification of flattening and ploughing mechanisms under conditions of high contact forces. Then, at high contact pressures, the share of mechanical cooperation of the surface asperities in the total resistance to friction increases.

A scatter plot between observed values and predicted values of roughness parameter Sa is shown in Figure 12. The values present normal distribution, and they are proportionally distributed along the diagonal line. In a normal distribution, the frequency of occurrence of events with the average value of the examined feature is therefore the highest, or, in other words, the probability of such an event occurring is the highest. The frequency of an event occurring (the probability of occurrence) decreases according to the increase in the deviation of the random variable from its expected value.

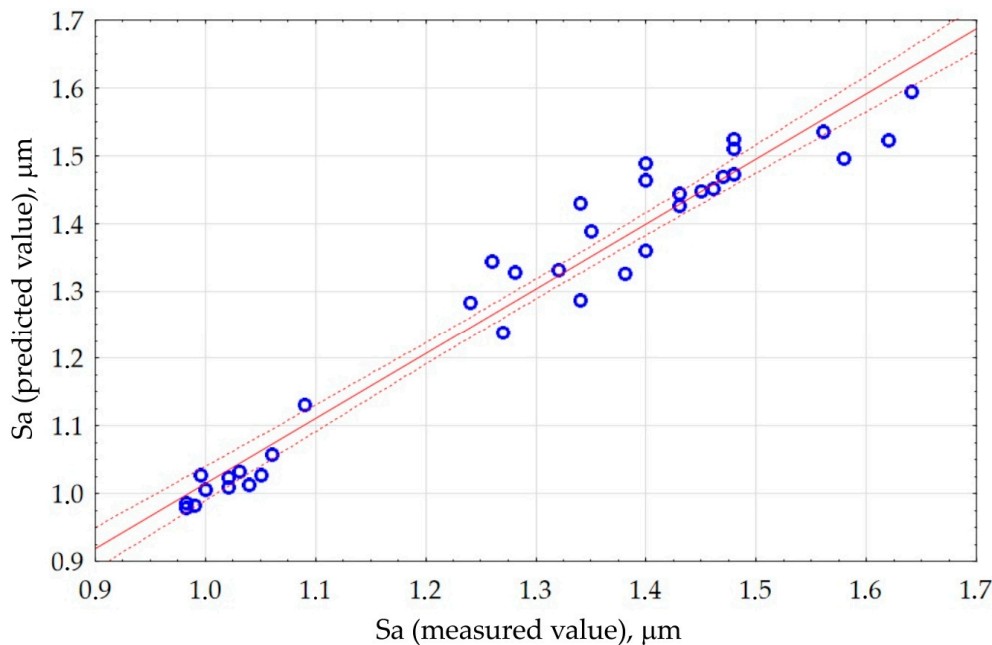

**Figure 12.** Scatter plot between observed values and predicted values of roughness parameter Sa using MLP 10-7-1.

### 3.3. MLP for Prediction of Skewness, Ssk

The MLP 10-6-1 neural network (Table 8) predicting the value of skewness, Ssk, of the sheet surface is characterised by worse Pearson's coefficients compared to the network predicting the value of the roughness parameter, Sa. The Pearson's coefficient values for the training, test and validation sets are 0.655, 0.298 and 0.844, respectively. The value of the error ratio for individual input variables (Table 14) is much more even compared to the network predicting the value of the Sa parameter (Table 13). However, none of the quotient values is lower than 1. Oil pressure, average value of contact force and oil viscosity have the greatest informative capacity.

**Table 14.** Sensitivity analysis of input variables for MLP 10-6-1.

| Oil Pressure | $F_{Nmean}$ | Oil Viscosity | $F_{NSD}$ | $F_{Nsk}$ | $F_{Nku}$ |
|---|---|---|---|---|---|
| 1.402 | 1.180 | 1.112 | 1.015 | 1.008 | 1.000 |

The experimental results and the response surfaces of the MLP 10-6-1 network of the influence of the kurtosis of contact force and the lubrication pressure on the value of the skewness, Ssk, are shown in Figure 13a,b, respectively. The prediction of the neural network is a flattening of the experimental curve. Due to the learning algorithm being stopped after the minimum error value for the test set is reached, the neural network reaches the optimal quality of data generalisation. Skewness, Ssk, increases with increasing lubrication pressure. Most of the sheets tested experimentally are characterised by a positive skewness, Ssk, value (Figure 13). Intense friction leads to rough interaction between the surface asperities, because of which, the surface is flattened.

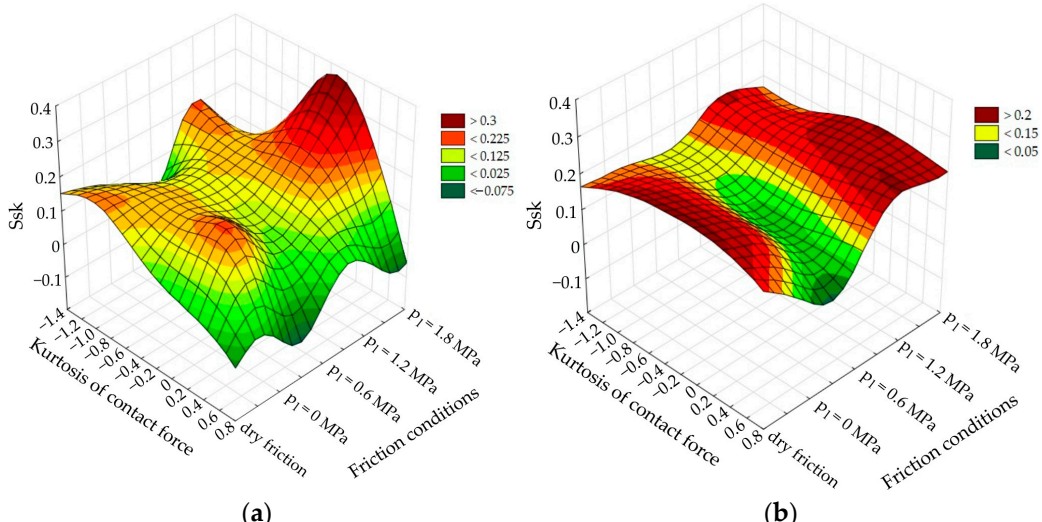

**Figure 13.** The effect of kurtosis of contact force and oil pressure on the value of the Ssk parameter: (**a**) measured data, (**b**) response surface for predicted data.

The influence of oil pressure and lubricant viscosity on the value of the roughness parameter, Ssk, is represented by the saddle surface. For dry friction (kinematic viscosity 0 mm$^2$/s), the skewness, Ssk, value shows negative values. It is local in nature. So, the neural network model predicts a saddle surface in a more flattened form, not considering local disturbances of experimental data. In the absence of lubrication, the Ssk value takes relatively large values above 0.3 (Figure 14). Increasing the lubricant pressure under pressure-assisted lubrication slightly increases the skewness value, but this effect is valid for oil with a viscosity of about 600 mm$^2$/s, corresponding to the saddle point position. A saddle point is a point on a surface with the property that, in any of its surroundings, there are points lying on both sides of the tangent.

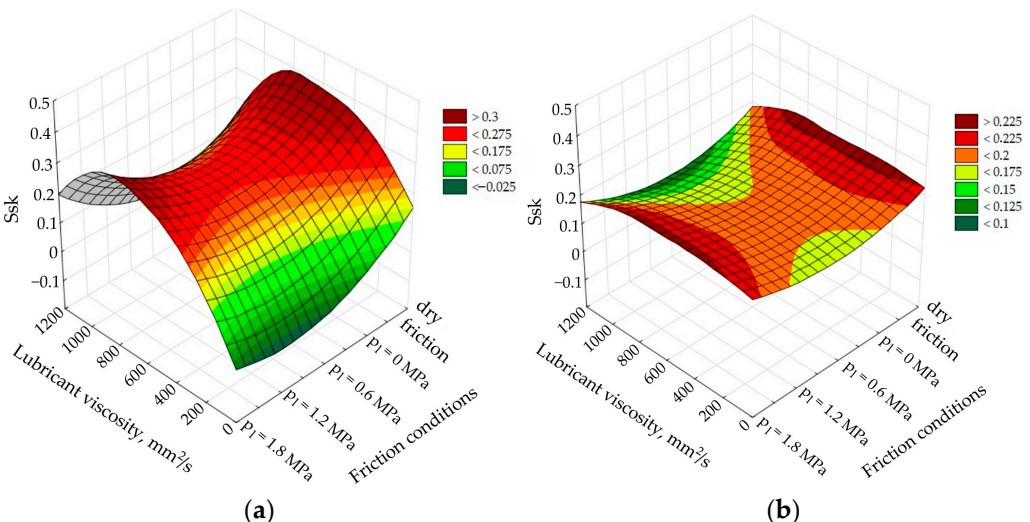

**Figure 14.** The effect of oil viscosity and lubricant pressure on the value of the Ssk parameter: (**a**) measured data, (**b**) response surface for predicted data.

In general, the shape of the predicted response surface for the network predicting the skewness, Ssk, value deviates much more from the experimental surfaces than the network predicting the value of average roughness Sa (Section 3.2). The reasons should be grounded in different values of the activation function of neurons. The function outputs, y, of the MLP analysed in this section are exponential, and they are in the range of 0 to ∞. The optimal MLP for modelling the average roughness, Sa, contains neurons with hyperbolic

activation function with the output values in the range between −1 and +1. So, owing the possibility of including both negative and positive values of the output, the MLP 10-7-1 (Sa) network tends to better match the experimental data than the MLP 10-6-1 (Ssk).

### 3.4. MLP for Prediction of Kurtosis, Sku

The architecture of the ANN that provides the smallest error values for the training and test sets is the MLP 10-9-1 network containing neurons with the tanh activation function (Table 12). For the training set, the Pearson's correlation is 0.993, and for the test and validation sets, 0.954 and 0.837, respectively. The average value of contact force and oil pressure at almost the same level affects the value of kurtosis, Sku, of the sheet surface. In order of decreasing importance, kurtosis of contact force, standard deviation of contact force, oil viscosity and skewness of contact force were ranked further (Table 15).

**Table 15.** Sensitivity analysis of input variables for MLP 10-9-1.

| $F_{Nmean}$ | Oil Pressure | $F_{Nku}$ | $F_{NSD}$ | Oil Viscosity | $F_{Nsk}$ |
|---|---|---|---|---|---|
| 13.215 | 12.277 | 3.045 | 2.526 | 2.152 | 1.702 |

The influence of oil viscosity and lubrication pressure on the value of the kurtosis, Sku, is given by a typical saddle curve (Figure 15). In the absence of lubrication and when lubricating with oil with a viscosity of 1.136 mm$^2$/s, the skewness values are below 3. Under these conditions, the distribution curve is platykurtoic with relatively low valleys and few high peaks. In the second case (Sku > 3), the distribution curve of the profile is leptokurtoic and is characterised by relatively low valleys and many high peaks [42]. The roughness parameter, Sku, increases to values above 3 with the increase in the lubricant pressure. This is a result of the formation of a lubricant 'cushion' and the separation of rubbing surfaces.

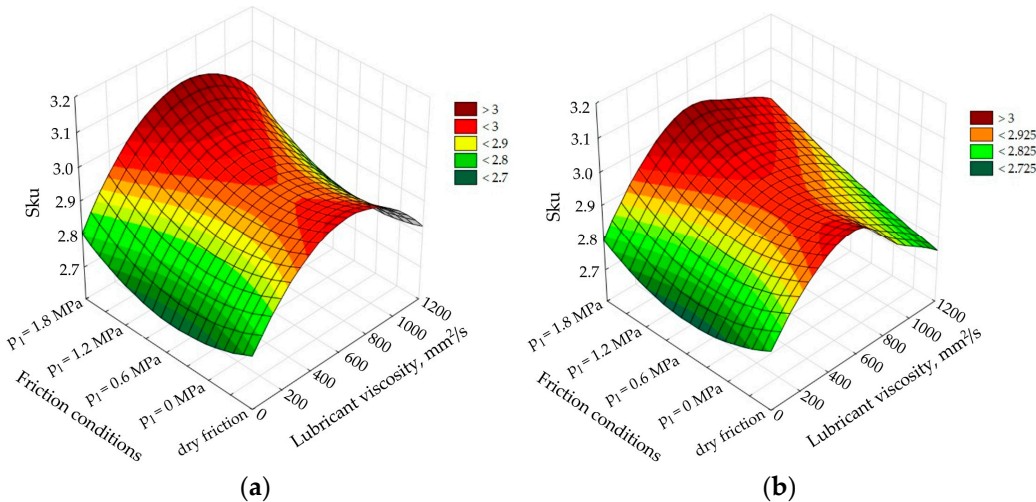

(**a**)                                        (**b**)

**Figure 15.** The effect of lubricant pressure and kurtosis of normal force on the value of the Ssk parameter: (**a**) measured data, (**b**) response surface for predicted data.

Increasing the average signal of contact force clearly reduces the value of kurtosis, Sku (Figure 16). This effect is observed for the entire range of oil viscosity changes. Under dry friction conditions (viscosity 0 mm$^2$/s), the distribution curve of the surface profile is platykurtoic with high peaks or deep scratches, which are formed as a result of the mechanical action of the summits of asperities under lubricant-free conditions.

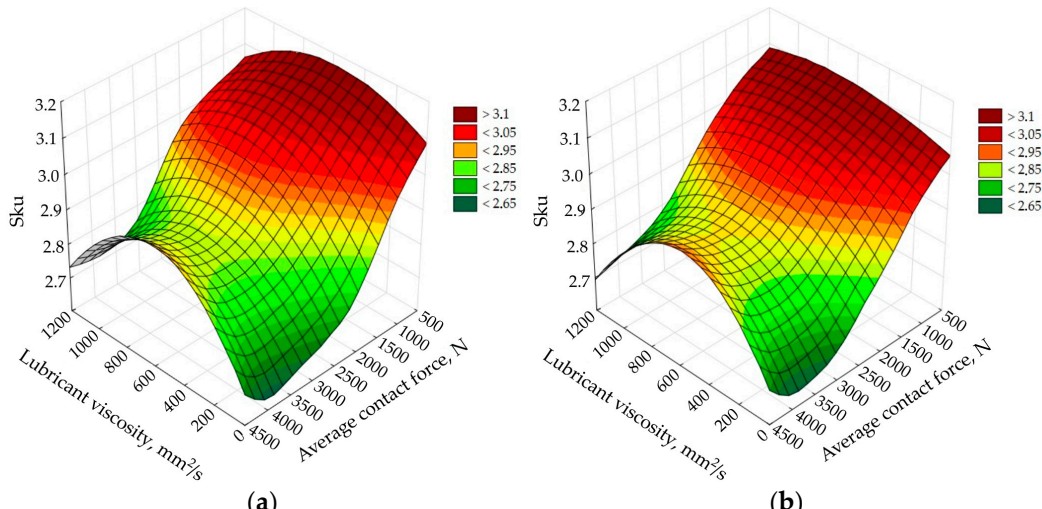

**Figure 16.** The effect of kurtosis of normal force and oil pressure on the value of the Sku parameter: (**a**) measured data, (**b**) response surface for predicted data.

Changing the friction conditions from dry friction to oil lubrication leads to an increase in the kurtosis, Sku, value (Figure 17). Similarly, under specific friction conditions, an increase in the average signal of contact force causes a decrease in the Sku parameter. As mentioned earlier, the lubrication of the contact interface, under pressure-assisted lubrication conditions, limits the mechanical interaction of the friction surfaces, specifically the interaction of the hard tool with the relatively soft sheet metal. An additional mechanism that is activated under high contact forces is the phenomenon of strain hardening. This phenomenon consists of an increase in the yield stresses that plasticise the material of the surface asperities along with the increasing plastic deformation of the sheet material.

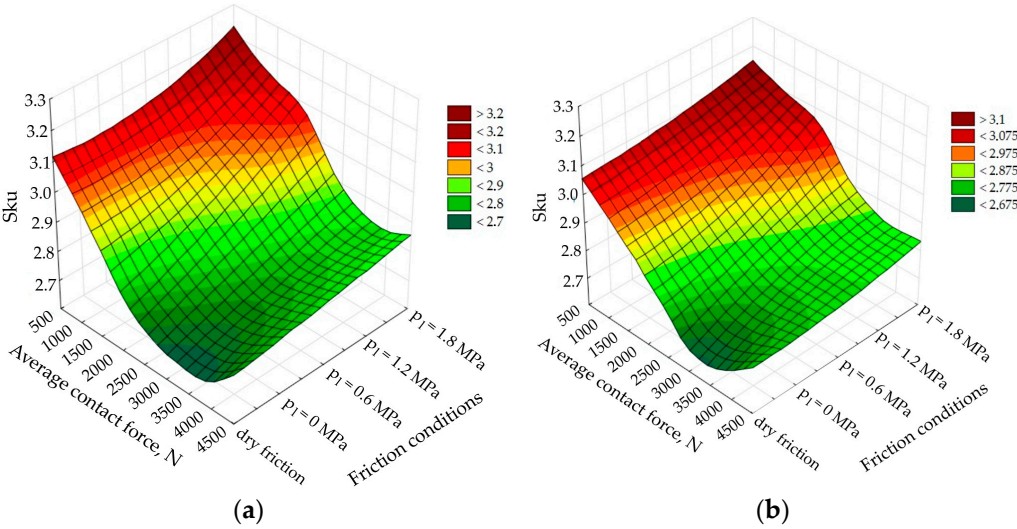

**Figure 17.** The effect of lubricant pressure and kurtosis of normal force on the value of the Sku parameter: (**a**) measured data, (**b**) response surface for predicted data.

The influence of lubricant viscosity on the coefficient of friction is closely related to the topography of the sheet metal surface [43]. The application of oil with high viscosity on the surface leads to the decrease in the coefficient of friction [44]. Lee et al. [45] found that high-viscosity lubricants provided lower friction coefficient values. Increasing the contact pressure results in intensified flattening of the summits of the surface asperities [39], and as a result, the value of the Sku parameter decreases with the increase in contact force. The increase in the pressure of the oil contained in closed lubricant pockets limits the

mechanical interaction of the surface asperities and reduces friction by creating a pressure lubricant cushion [8]. The largest change in the Sku parameter occurs in conditions of a lack of pressurised lubrication and dry friction [38].

Detailed prediction statistics for the MLP 10-7-1 networks used for the prediction of roughness parameters Sa, Ssk and Sku are listed in Table A1 (Appendix A).

Since a neural network is nothing more than a multidimensional approximation mechanism, its use for this task is probably redundant. The surfaces obtained in the experiment (Figures 8a, 9a, 10a, 11a, 13a, 14a, 15a, 16a and 17a) and as a result of the prediction of the ANN (Figures 8b, 9b, 10b, 11b, 13b, 14b, 15b, 16b and 17b) are quite smooth, and it is most likely that the usual approximation would give a similar result.

## 4. Conclusions

The MLPs were used to explore the effect of friction process parameters on the value of selected parameters of the surface roughness (Sa, Ssk and Sku). The main results are as follows:

- The lubricant pressure and lubricant viscosity are variables that most significantly affect the value of the roughness parameter, Sa, of sheet metal after the friction process.
- Under lubricated conditions, if the lubricant pressure increases, the Sa value decreases. This generally applies to negative values of the kurtosis of contact force.
- The most unfavourable friction conditions due to the minimisation of the Sa roughness parameter occur at the highest average contact force and in dry friction conditions.
- Oil pressure and oil viscosity have the greatest impact on the skewness, Ssk, of the sheet metal surface after the friction process.
- Skewness, Ssk, increases with increasing lubrication pressure. After the friction process, most samples are characterised by a positive skewness, Ssk, value.
- The average value of contact force and oil pressure at almost the same level affects the kurtosis, Sku, value of the sheet surface.
- Changing the friction conditions from dry friction to oil lubrication leads to an increase in the kurtosis, Sku, value.

**Author Contributions:** Conceptualization, T.T., K.S. and M.S.; methodology, T.T., K.S. and M.S.; validation, T.T., K.S. and M.S.; investigation, T.T., K.S. and M.S.; data curation, T.T., K.S. and M.S.; writing—original draft preparation, T.T. and K.S.; writing—review and editing, T.T. All authors have read and agreed to the published version of the manuscript.

**Funding:** This research received no external funding.

**Institutional Review Board Statement:** Not applicable.

**Informed Consent Statement:** Not applicable.

**Data Availability Statement:** Data is contained within the article.

**Conflicts of Interest:** The authors declare no conflict of interest.

## Appendix A

**Table A1.** Detailed qualitative statistics of MLP networks used to predict Sa, Ssk and Sku roughness parameters.

| Parameter | Set | MLP 10-7-1 (Sa) | MLP 10-6-1 (Ssk) | MLP 10-9-1 (Sku) |
|---|---|---|---|---|
| Minimum predicted value | train | 0.97985 | −0.0136 | 2.64590 |
| | test | 0.98254 | 0.14685 | 2.69304 |
| | validation | 1.28784 | 0.10030 | 2.82405 |

**Table A1.** *Cont.*

| Parameter | Set | MLP 10-7-1 (Sa) | MLP 10-6-1 (Ssk) | MLP 10-9-1 (Sku) |
|---|---|---|---|---|
| Maximum predicted value | train | 1.53516 | 0.26168 | 3.14747 |
| | test | 1.46391 | 0.25249 | 3.08723 |
| | validation | 1.59470 | 0.20856 | 3.10607 |
| Minimum value of residuals | train | −0.08946 | −0.14348 | −0.03739 |
| | test | −0.08372 | −0.18557 | −0.08319 |
| | validation | −0.04739 | −0.14210 | −0.13022 |
| Maximum value of residuals | train | 0.09547 | 0.16149 | 0.05049 |
| | test | 0.02304 | 0.30841 | 0.10875 |
| | validation | 0.05216 | 0.13344 | 0.06119 |
| Minimum value of standardized residuals | train | −3.01634 | −2.31434 | −3.28748 |
| | test | −2.34724 | −1.44777 | −1.51255 |
| | validation | −1.61685 | −2.03700 | −2.42133 |
| Maximum value of standardized residuals | train | 3.21883 | 2.60478 | 4.43908 |
| | test | 0.064595 | 2.40618 | 1.97730 |
| | validation | 1.77976 | 1.91295 | 1.13772 |

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
