# Peer review of "Analysis of Surface Topography Changes during Friction Testing in Cold Metal Forming of DC03 Steel Samples"

_coatings, doi:10.3390/coatings13101738_

Round 1

Reviewer 1 Report

 major revision

major revision 

Author Response

Dear reviewer
We would like to thank the reviewers for careful and thorough reading of this manuscript and for the thoughtful comments and constructive suggestions, which helped us to improve the quality of this manuscript. 

Reviewer 2 Report

In this paper one of the artificial intelligence tools, namely artificial neural networks, was employed to find the relation between parameters of the friction process and the main surface roughness parameters. The article has scientific novelty and practical significance. It will be useful to other researchers.

The positive points of the article are:

1. Analysis of publications made on this topic, which contains 26 studies, most of which are in the last 5 years.

2. Very detailed and with high-quality graphic drawings and photos describes the research methodology and the equipment on which the research was carried out.

3. Scientifically substantiated and with detailed discussions, a technique for constructing a neural network model is described.

4. The convergence of the neural network model and experimental data is high. Which is shown in Figures 8-17.

5. The conclusions drawn from the research results are formulated concisely and reflect the essence of the research.

I recommend the article for publication in this version, and I express my gratitude to the authors for the high level of research and the high quality of the presentation of the results.

Author Response

(The authors gave the same response as above.)

Reviewer 3 Report

The manuscript entitled Analysis of surface topography changes during friction in cold metal forming of DC03 steel samples is very interesting  and overall the work is good.

However, in my opinion the topic is not suitable for Coatings. But, since this aspect is not part of my reviewer role, without taking into account this ascpect,  I recommend acceptance of the manuscript with some small adjustments:

- using the same type of notation, with comma or period, for all the values (for example hardness 197.2HV at line 152, while at line 158  we have 1,136 mm2/s)

- double-check for spelling

Author Response

(The authors gave the same response as above.)

Reviewer 4 Report

The authors have provided a fairly large overview. How does the information from the papers in the review (for example, [26], which is extremely important) fit in with the results obtained by the authors?

Is it possible to obtain any results described in the review from the results obtained by the authors?

The authors go on to describe the structure of the neural network, but it is not clear exactly how the input values were formed and how the output value was arranged. The use of neural networks in technology is not yet generally accepted, so it should be described in detail.

Neural networks require large amounts of data to be used. It is not clear how much data was used for the analysis. Graph 12 shows that there are few such values.

Since the neural network is nothing more than a multidimensional approximation mechanism, its use for this task is probably redundant. The surfaces obtained in the experiment and as a result of the prediction of the neural network are quite smooth, most likely the usual approximation would give a similar result. It is not clear what advantages the use of a neural network has.

Most of the conclusions of the article are obvious.

As you know, the viscosity of liquids (lubricants) is very dependent on temperature, the article does not say anything about the effect of temperature, and changes in temperature should devalue all the data obtained.

Author Response

(The authors gave the same response as above.)

Round 2

Reviewer 1 Report

ACCEPTED PAPER FOR PUBLICATION 

accepted paper 

Reviewer 4 Report

The article may be published